# Late Bone Marrow Mononuclear Cell Transplantation in Rats with Sciatic Nerve Crush: Analysis of a Potential Therapeutic Time Window

**DOI:** 10.3390/ijms252312482

**Published:** 2024-11-21

**Authors:** Vanina Usach, Mailin Casadei, Gonzalo Piñero, Marianela Vence, Paula Soto, Alicia Cueto, Pablo Rodolfo Brumovsky, Clara Patricia Setton-Avruj

**Affiliations:** 1Cátedra de Química Biológica Patológica, Departamento de Química Biológica, Facultad de Farmacia y Bioquímica, Universidad de Buenos Aires, Buenos Aires C1113AAD, Argentina; vaninausach@gmail.com (V.U.); gonzalopiniero@gmail.com (G.P.); paula.asoto02@gmail.com (P.S.); 2Instituto de Química y Fisicoquímica Biológicas (IQUIFIB), Universidad de Buenos Aires-Consejo Nacional de Investigaciones Científicas y Técnicas (CONICET), Buenos Aires C1113AAD, Argentina; vencemarianela@yahoo.com.ar; 3Instituto de Investigaciones en Medicina Traslacional (IIMT), Consejo Nacional de Investigaciones Científicas y Técnicas (CONICET), Universidad Austral, Pilar B1629AHJ, Argentina; maicasadei@hotmail.com (M.C.); pbrumovs@austral.edu.ar (P.R.B.); 4Facultad de Ciencias Biomédicas, Universidad Austral, Pilar B1629AHJ, Argentina; 5Servicio de Neurología, Hospital Español de Buenos Aires, Buenos Aires C1209, Argentina; alicueto@gmail.com

**Keywords:** bone marrow mononuclear cells, sciatic nerve crush, transplantation, neuropathic pain, regeneration

## Abstract

After peripheral nerve injury, axon and myelin regeneration are key events for optimal clinical improvements. We have previously shown that early bone marrow mononuclear cell (BMMC) transplantation exerts beneficial effects on myelin regeneration. In the present study, we analyze whether there is a temporal window in which BMMCs migrate more efficiently to damaged nerves while still retaining their positive effects. Adult Wistar rats of both sexes, with sciatic nerve crush, were systemically transplanted with BMMC at different days post injury. Vehicle-treated, naïve, and sham rats were also included. Morphological, functional, and behavioral analyses were performed in nerves from each experimental group at different survival times. BMMC transplantation between 0 and 7 days after injury resulted in the largest number of nested cells within the injured sciatic nerve, which supports the therapeutic value of BMMC administration within the first week after injury. Most importantly, later BMMC administration 7 days after sciatic nerve crush was associated with neuropathic pain reversion, improved morphological appearance of the damaged nerves, and a tendency toward faster recovery in the sciatic functional index and electrophysiological parameters. Our results thus support the notion that even delayed BMMC treatment may represent a promising therapeutic strategy for peripheral nerve injuries.

## 1. Introduction

Peripheral nerve injury is typically associated with Wallerian degeneration, both at and distal from the site of injury [1]. Various types of infiltrating immune cells participate in debris removal [2,3] and clearance of partially damaged axons [4], causing variable alterations in limb sensitivity, functional loss, and neuropathic pain [5,6]. Because axonal regeneration and re-myelination are key events for optimal clinical improvement, finding relevant therapeutic strategies is of great importance.

Stem cell therapy has emerged as a promising approach in the field of regenerative medicine [7,8]. In particular, bone marrow mononuclear cells (BMMCs) have appeared as a suitable option because of their convenient isolation protocol, high yield and survival rate after transplantation, and low immunogenicity [9]. Their therapeutic potential has been addressed in several animal models of injuries [10,11,12,13,14,15,16]. BMMCs are capable of stimulating neovascularization [17] or re-epithelialization and wound healing [18], and their therapeutic value has been further studied in several recent clinical trails [19,20,21,22,23,24,25,26]. We have previously shown in rat models of irreversible [27] and reversible Wallerian degeneration [8,28] that BMMCs migrate almost exclusively to the injured site. Interestingly, the administration of BMMCs immediately after sciatic nerve crush promotes considerable regeneration in nerve and myelin, partial recovery of electrophysiological properties, and the complete prevention of associated neuropathic pain [8].

However, the question remains whether there is an optimal temporal window for BMMC administration after injury. In animal models of myocardial infarction, it has been shown that the most beneficial results are obtained when transplantation occurs during the acute phase [11,29]. Studies in humans also point to a temporal window when BMMC transplant is most effective, as early interventions in patients suffering traumatic paraplegia resulted in better clinical outcomes [25]. Of note, a time window of therapeutic opportunity for the use of BMMC in animals undergoing peripheral nerve injury has not been established yet.

Here, we aimed at determining the temporal window in which transplanted BMMC migrate most efficiently in a model of sciatic nerve crush, and whether this administration results in similar behavioral, morphological, and physiological improvements as those previously observed [8]. The results obtained in the present manuscript support the notion that even delayed BMMC treatment may represent a promising therapeutic strategy for peripheral nerve injuries.

## 2. Results

### 2.1. Kinetics of BMMC Migration

We observed the largest number of nested ^EGFP^BMMCs in the distal area of the ipsilateral (IL) nerve in rats transplanted between 0 and 7 dpi, with significant differences in naïve rats, but not between early and late transplant (Figure 1a,b). Transplant 14 dpi and onward showed a marked decrease in the number of recruited cells, which resulted in no significant differences from naïve rats (Figure 1a). Virtually no ^EGFP^BMMC were detected in naïve, contralateral (CL) nerves or the proximal stump of IL nerves (Figure 1a,b).

Considering the results obtained, we chose 7 dpi transplant to address the effects of late BMMC administration on the function and morphology of injured nerves.

### 2.2. Effects of Late BMMC Transplant on Mechanical and Coldallodynia

All injured rats exhibited a progressive decrease in IL mechanical withdrawal thresholds and an increase in cold withdrawal frequency and score during the first 7 dpi (Figure 2). Vehicle-treated animals maintained IL mechanical allodynia until 21 dpi, followed by a slow recovery that was only complete at 35 dpi. In contrast, late BMMC-treated rats showed a progressive increase in IL mechanical withdrawal threshold starting at 8 dpi and considerable improvement during the first week after transplant, reaching basal withdrawal levels at 21 dpi and onward (Figure 2a).

Analysis of cold allodynia showed that vehicle-treated rats maintained significant increases in IL cold withdrawal frequency (Figure 2b top panel) and score (Figure 2b bottom panel), both returning to basal levels at 35 dpi. In contrast, late BMMC transplant resulted in a sharp reduction in IL cold allodynia over the first week after transplant, with animals exhibiting basal cold withdrawal frequency (Figure 2b top panel) and score (Figure 2b bottom panel) as soon as 21 dpi.

Neither vehicle-treated nor late BMMC-treated rats exhibited noticeable changes in CL mechanical withdrawal threshold or cold withdrawal frequency and score throughout the study (Figure 2).

### 2.3. Effects of Late BMMC Treatment on SFI, Distal Latency, and CMAP

Both experimental groups exhibited similarly altered footprints at 7 and 14 dpi; none of the toes were evident (Figure 3a right top panel), and a significant reduction was observed in SFI and footprint area (Figure 3a bottom panel). At 21 dpi, vehicle-treated rats showed undifferentiated 2nd–4th toe prints and a significant decrease in SFI compared to naïve and BMMC-treated rats (Figure 3a) and in footprint area compared to naïve rats (Figure 3a bottom panel). At 28 dpi, all toes showed a separate print and the SFI was not significantly different from naïve rats (Figure 3a bottom panel); however, the hind paw area was smaller than that observed in naïve animals. In contrast, a clear print was observed for all toes in BMMC-treated rats starting 21 dpi (Figure 3a right top panel), and the SFI and hind paw area (Figure 3a bottom panel) exhibited values similar to naïve rats.

Distal latency showed a significant increase in both experimental groups from 7 dpi and up to 28 dpi, with no statistically significant differences between them (Figure 3b). However, a tendency toward a faster but not significant recovery was noticed in late BMMC-transplanted rats at 35 dpi. 

Analysis of CMAP amplitude in vehicle-treated rats showed a reduction in amplitude values from 7 to 21 dpi, followed by a partial recovery at 28 dpi and full recovery with basal values at 35 dpi (Figure 3b). BMMC-treated rats exhibited a similar CMAP amplitude change as compared to vehicle-treated rats, although a significant increase was only detected from 21 dpi (Figure 3b).

### 2.4. Effects of Late BMMC Transplant on Axon Number and Re-Myelination State

The analysis of semi-thin sections of sciatic nerves from naïve animals showed healthy axons, most of them of large caliber (Figure 4a.1 arrows). Conversely, analysis at 7 dpi revealed axon and myelin debris characteristic of Wallerian degeneration (Figure 4a.2 asterisk). As survival time proceeded, vehicle-treated animals 14 dpi exhibited axon and myelin debris (Figure 4b, asterisk) and newly synthesized axons of small caliber (Figure 4b, arrowhead). Studies at 21 dpi showed no myelin or axon debris but small newly formed axons (Figure 4c). Finally, most axons resembled those of naïve rats at 28 dpi, although with fewer myelin layers (Figure 4d, arrow). In contrast, the BMMC-treated group barely showed myelin or axon debris 14 dpi (Figure 4e, asterisk) but revealed a large number of newly synthesized axons of small caliber (Figure 4e, arrowhead). Beginning at 21 dpi, axon caliber increased and reached a naïve-like phenotype (Figure 4g, arrows).

Ultrastructural analyses of sciatic nerves (Figure 5) showed mostly myelinated axons of large caliber in naïve animals (Figure 5a.1, arrows). Seven and fourteen dpi, vehicle-treated animals only showed degenerating axons and myelin debris (Figure 5a.2,b, arrow heads and asterisk, respectively). Myelin and axon debris disappeared at 21 dpi, and newly formed myelinated axons were observed of large caliber and unusual shape (Figure 5c). As spontaneous regeneration proceeded, axons of large caliber with a thin myelin layer and naïve-like shape were observed at 28 dpi (Figure 5d, arrow). In turn, the BMMC-treated group showed some myelinated axons of large caliber wrapped by a thin myelin layer at 14 dpi (Figure 5e, arrow heads). Axons became more myelinated at 21 dpi and acquired a normal myelin sheath and shape at 28 dpi (Figure 5g, arrows).

As regards axon quantification, the results in Figure 6 show a sharp decrease in the total number of axons from 7 dpi in the vehicle-treated group (15.32 ± 6.4%) and a return to normal values at 28 dpi (102 ± 14.8%). In the BMMC-treated group, the total number of axons began to increase as from 7 dpi—the day of BMMC transplant—and reached values non-significantly different from naïve nerves at 21 dpi (86 ± 3.46%).

Digging deeper into axon number and re-myelination, we found a sharp decrease in the number of large-caliber myelinated axons 7 dpi in the vehicle-treated group (16.6 ± 10.5%) and a return to normal values 28 dpi (95 ± 5%). In contrast, the BMMC-treated group evidenced a marked recovery in the number of large-caliber myelinated axons as from 14 dpi (74.66 ± 3.28%), with values non-significantly different from naïve nerves. Small-caliber axons increased in number in the vehicle-treated group at 7 dpi (100%) and gradually decreased to normal values at 28 dpi (5 ± 5%). In contrast, the BMMC-treated group revealed a sharp decrease in the number of small-caliber axons from 14 dpi (25.33 ± 3.28%), once again with values non-significantly different from naïve nerves.

Regarding the g-ratio (Figure 6b), an increase was observed in vehicle-treated animals 7 dpi as compared to naïve animals (0.75 ± 0.08). The g-ratio remained high, up to 28 dpi in the vehicle-treated group, but recovered normal values from 21 dpi in the BMMC-treated group (0.60 ± 0.1). The number of myelin layers per 100 nm decreased at 7 dpi in vehicle-treated rats (8.69 ± 0.58) and remained significantly different from naïve nerves until 28 dpi (9.98 ± 0.78); in the BMMC-treated group, no significant differences were observed from naïve nerves from 14 dpi (9.76 ± 0.52). Finally, the intraperiod line revealed significant differences only at 14 and 21 dpi (10.19 ± 1.4 and 9.73 ± 1.4) in sciatic nerves from animals in the vehicle-treated group.

### 2.5. Effects of Late BMMC Transplant on MBP and βIII-Tubulin Protein Levels and Distribution

Blotting analysis revealed a clear decrease in MBP levels in the distal stump of vehicle-treated rats (Figure 7a), with considerably low protein levels at 28 dpi (Figure 7b). Of note, a much smaller decrease was observed in late BMMC-treated rats (Figure 7a,b). In fact, significant differences were observed between experimental groups at all time-points evaluated (Figure 7b). βIII-tubulin levels also showed a decrease 14 and 21 dpi in vehicle-treated rats, followed by a substantial recovery 28 dpi (Figure 7a,b). In contrast, late BMMC transplant partially prevented the decrease in βIII-tubulin levels, only rendering slight statistically significant differences at 14 and 21 dpi (Figure 7b).

Immunohistochemical analysis confirmed the immunoblotting results. Thus, while MBP-like immunoreactivity (LI) in naïve nerves always produced a continuous and homogenous signal with an organized number of nuclei, sciatic nerve crush resulted in clear signs of demyelination, with a profound decrease in MBP-LI and the appearance of MBP clusters 7 and 14 dpi (Figure 8 asterisks), and also a significant increase in total nuclei, which became disorganized. BMMC treatment at both time points improved MBP-LI signal and reduced the number of clusters as compared to the vehicle-treated group; however, these effects were sharper in early BMMC-treated rats. Moreover, late BMMC-treated rats showed signs of improvement in MBP-LI as early as 14 dpi with the presence of MBP clusters accompanied with neosynthesized fibers (Figure 8 hashtag), and onward accompanied by a reduction in the number of total nuclei and recovery of their organization (Figure 8 and 10a). The effect of late BMMC treatment on the demyelination process was confirmed through p75NTR-LI, a marker of repairing Büngner SC.

Finally, immunohistochemical analysis of axonal marker βIII-tubulin (Figure 9 and Figure 10b) revealed that, compared to the homogenous signal observed in naïve rats, vehicle-treated rats exhibited a considerable decrease in βIII-tubulin-LI up to 14 dpi, as well as the appearance of βIII-tubulin clusters (Figure 9 asterisks). In contrast, late BMMC-treated rats exhibited a noticeably smaller drop in βIII-tubulin-LI and cluster formation, with signs of significant improvement from 14 dpi (Figure 9 hashtags). The quantity and organization of total nuclei had the same pattern as the one described in Figure 8.

## 3. Discussion

The present study supports the therapeutic value of systemic BMMC transplantation at a time when peripheral neuropathy is already installed. Transplant within the first 7 dpi in rats resulted in the most efficient cell recruitment, while later injections showed considerable reductions in BMMC infiltration. These observations are in line with the peak in total nuclei, indicating high Schwann cell (SC) proliferation and a rise in infiltrating inflammatory cells, both hallmarks of the Wallerian degeneration process [30,31,32]. BMMC administration 7 dpi efficiently reduced mechanical and cold allodynia and induced a faster recovery in the total number of axons, their myelination state, and the levels and organization of MBP and βIII-tubulin. A tendency toward accelerated functional recovery was also observed from 14 dpi in terms of CMAP amplitude and SFI.

Analyses establishing the temporal window with the highest therapeutic impact of BMMC transplantation after injury remain scarce. Several authors have demonstrated beneficial effects in intravenous BMMC transplant models with transplant occurring between 1 h and 7 dpi [33,34,35]; however, no beneficial effects were obtained when transplantation occurred at 14 or 30 dpi [35]. Although fast therapeutic actions are key in inducing recovery, safe and beneficial effects have also been shown in dogs with chronic spinal cord injury [36,37] and rats with chronic stress [38] and depression [39].

Upon injury, SCs actively proliferate, acquire a repair phenotype [40,41], and secrete trophic factors and cytokines that promote neuronal survival and leukocyte influx [31]. This microenvironment facilitates the signaling, migration and nesting of a variety of immune cells [42,43,44], and the activation of resident mesenchymal precursors [45] or dedifferentiated SC with a mesenchymal cell-like phenotype [46]. This pro-regenerative, inflammatory microenvironment present in the sub-acute phase is essential to the recruitment of endogenous immune cells [47] and transplanted BMMC [48,49]. No significant migration was observed in the spontaneous regeneration phase, which suggests a much lesser chemoattractant environment as pro-inflammatory signals decrease.

Worth mentioning is that late BMMC transplantation was highly efficient in reducing and even eliminating pain-like behavior in injured rats. This finding agrees with previously published studies demonstrating that unilateral injection of BMMC into several muscles in rats with diabetic neuropathy ameliorates mechanical and cold hyperalgesia [50] and that early intravenous administration of BMMC completely prevents mechanical allodynia after sciatic nerve crush [8]. However, in mice with lumbar 5 spinal nerve injury, the intrathecal injection of BMMC 1 dpi resulted in a slight, although significant, increase in IL mechanical withdrawal threshold [51].

The mechanisms by which BMMCs modulate neuropathic pain are yet to be determined. BMMCs represent a heterogeneous population including MSC, hematopoietic stem cells, endothelial cells, and hematopoietic cell precursors [52] which, according to Song et al. [53], may surpass MSC neuroregenerative abilities, associated with the variety of cells comprising this population and their synergistic interactions [53,54]. Remarkably, only a small percentage of transplanted cells appear to be integrated in the injured tissue, where they undergo transdifferentiation to Schwann cells, macrophages, and axonal cells [28,29], making it difficult to evaluate through in vivo characterization their original phenotype and conclude which subpopulation is recruited in the injured nerve, exerting their beneficial effects. The production of several cytokines [55] and the release of factors which can mediate potent antioxidant effects [56] have also been proposed as a potential mechanism of action. Interestingly, it has been determined that the upregulation of IL-6, IL-1β, and TNF-α, among others [57,58], strongly influences neuropathic pain. It may be thus speculated that BMMCs relieve neuropathic pain through an immunomodulatory action, previously described by our group, suppressing the expression of several pro-inflammatory cytokines and inducing anti-inflammatory cytokines and macrophage phenotype changes [51,59]. Taking into consideration our previous results on cytokine expression, and other authors’ research that demonstrated that some chemokines and neuroinflammation signaling are only upregulated in males in terms of neuropathic pain [60,61,62], in the present manuscript male rats were only evaluated in terms of neuropathic pain. Thus, one limitation in our study is the lack of data in female rats. Several studies over recent years have exposed and analyzed a variety of sex-related differences in the physiopathology of chronic pain [63,64,65,66,67]. Therefore, it remains to be determined if BMMC treatment is equally effective in reducing mechanical allodynia, or if the temporal therapeutic window will be similar in female rats than what has been presented here in male rats.

The SFI provides valuable information concerning the recovery of sensory-motor connections and cortical integration related to gait function after peripheral nerve injury [68]. Previous reports established a clear correlation between the SFI, histomorphometric measurements and electrophysiological analysis [69,70]. However, our results do not actually support such a correlation, as also previously reported [71]. This discrepancy may be due to the severe nerve degeneration observed in the first 14 dpi and the difficulty in obtaining a reliable paw print [72]. Here, morphometric aspects showed an early response to treatment—evidenced by a recovery in axon number and myelin quantity and organization—, while improvements in functional properties were detected at later time points. Worth highlighting is that BMMC-treated rats exhibited SFI recovery 1 week earlier than vehicle-treated ones and a tendency toward accelerated recovery in CMAP amplitude, reflecting some correlation between histomorphometric and functional parameters.

Our results are also in line with clinical trials demonstrating the therapeutic value of BMMC. Patients with diabetic neuropathy have shown ameliorated pain and functional recovery in peripheral nerves, accompanied by increased nerve blood flow [50]. Chronic lymphedema patients revealed an improvement in limb circumference and walking abilities [25]. Additionally, children with cerebral palsy improved their gross motor function and spasticity 6 months after transplantation [73]. Finally, patients with spinal cord injury or chronic traumatic brain injury [49] receiving BMMC treatment exhibited significantly reduced functional deficits and became more independent in everyday tasks [48].

## 4. Materials and Methods

### 4.1. Animals

All animals were treated humanely, and experimental procedures, including number of animals in each experiments, were performed following the guidelines of Comité de Bioética at Facultad de Farmacia y Bioquímica, Universidad de Buenos Aires (CICUAL 1488/19), the Institutional Animal Care and Use Committee (IACUC; 17-04) of the IIMT CONICET-Universidad Austral, and in accordance with the NIH Guide for the Care and Use of Laboratory Animals (NIH Publications 86-23), the Directive 2010/63/EU for animal experiments of the European Parliament and the Council of the European Union, taking in account the 3R principle (replace, reduce, and refine). Adult wild type Wistar rats (^WT^Wistar rats, 270–300 g) and transgenic background (enhanced green fluorescent protein-expressing ^EGFP^Wistar rats, 300–350 g) of either sex were used except in behavioral studies where male ^WT^Wistar rats were used. Animals were housed in a light- and temperature-controlled room with a 12-h-light/dark cycle and access to food and water ad libitum. The transgenic strain was generously provided by Dr Mathieu (IQUIFIB-CONICET, Buenos Aires, Argentina) and Dr Pitossi (Fundación Instituto Leloir, Buenos Aires, Argentina). 

### 4.2. Sciatic Nerve Crush

^WT^Wistar rats were anesthetized intraperitoneally with ketamine (75 mg/kg) and xylazine (10 mg/kg), and their right sciatic nerve was exposed and crushed for 8 s at mid-thigh level using jeweler’s forceps [8,28].

### 4.3. BMMC Isolation and In Vivo Transplant

BMMCs were isolated from femurs and tibias from ^WT^ or ^EGFP^Wistar rats, as previously described [8,30]. Briefly, the bone marrow was extruded with DMEM + 10% fetal calf serum (Cripion Biotecnologia Ltd., Buenos Aires, Argentina) and the aspirate was centrifuged through a Ficoll-Paque Plus (GE Healthcare#17-1440-02, Chicago, IL, USA) density gradient. The mononuclear cell fraction was used for further experiments [8,28,29].

Animals were transplanted at different days post injury (dpi; 0 (early treatment) or 3–28 days (late treatment)) with 1 × 10^7 EGFP^BMMC (for cell recruitment analysis) or ^WT^BMMC (for effect analysis) in 200 µL through the lateral tail vein using a 21 G needle [8,28]. Cell recruitment at the different survival times was analyzed using an Olympus FV1000 confocal microscope.

### 4.4. Experimental Groups

Animals were divided into vehicle-(DMEM) and BMMC-treated groups receiving treatment at different dpi and then sacrificed at different survival times (mentioned in each figure). Naïve and/or sham rats were also included. The number of animals used in each experiment is also indicated in each figure legend. All experimental procedures were undertaken by researchers that were blind to experimental design in order to avoid bias.

### 4.5. Behavioral Testing

Animals were first acclimatized to the chambers or walking pathway in a quiet room during daytime before surgery and at different dpi, to evaluate the following tests:

### 4.6. Mechanical Allodynia

The mechanical withdrawal threshold was assessed using von Frey filaments of different bending forces (Stoelting Inc., Wood Dale, IL, USA). The centers of the plantar surface of the IL and CL hind paws were mechanically stimulated, following the modified up-down method of Dixon [74], to establish the 50% withdrawal threshold. A paw withdrawal reflex obtained with 6.0 g force or less was considered an allodynic response.

### 4.7. Cold Allodynia

A drop of acetone was gently brought into contact with the plantar surface of both hind paws 5 times (once every 5 min) in each one [75]. For withdrawal frequency analysis, a positive response was scored as 1, and lack of withdrawal as 0. For cold sensitivity scoring, responses to acetone were graded using a previously described 4-point scale [76]. Cumulative scores were generated, with the minimum score being 0 and the maximum score being 15. Statistically significant increases in withdrawal frequency and cold sensitivity scoring in response to acetone were interpreted as cold allodynia.

### 4.8. Walking Track Analysis

Rats were placed in a walking pathway [77] and allowed to walk down the track after dipping their hind feet in non-toxic ink. At least 3 prints of each foot were obtained and used to calculate the sciatic functional index (SFI) and the area of the footprint (Figure 11a); a value of 0 indicates a completely healthy nerve, whereas a value of −100 indicates total impairment of sciatic function. The areas were outlined manually and calculated using ImageJ 1.52n (Figure 11b), and the IL/CL ratio was then determined.

### 4.9. Distal Latency and Compound Muscle Action Potential (CMAP) Recording

Both parameters were recorded in vehicle- and BMMC-treated rats at different dpi as previously described [8] using a portable electromyography instrument (Cadwell Wedge Sierra II, Cadwell Labs, Inc., Kennewick, WA, USA). Briefly, animals were anesthetized, and body temperature was maintained at 37 °C with a thermal blanket. Recording and ground electrodes were placed in the soleus muscle and the tail, respectively. The IL sciatic nerve was exposed, and the distal end was electrically stimulated with 30 mA; a slight twitching of the limb was considered positive stimuli. The same procedure was carried out in naïve and CL nerves. The distal latency and the amplitude of the compound action potential (CMAP) were recorded.

### 4.10. Electron and Optical Microscopy Analysis

Fourteen, twenty-one, and twenty-eight dpi, animals were anesthetized and perfused with 4% paraformaldehyde plus 2.5% glutaraldehyde in 0.1 M phosphate buffer, pH 7.4. The distal area of the ipsilateral nerve and a naïve sciatic nerve were dissected, and tissue was prepared as previously described [33]. Semi-thin tissue sections were mounted onto glass slides and dyed with 0.5% Toluidine blue in sodium carbonate 2.5% (*w*/*v*). Ultra-thin sections were collected on 300 mesh copper grids, dyed with 2% (*w*/*v*) aqueous uranyl, stained with Reynolds solution, and later analyzed in a Zeiss EM 109T electron microscope.

### 4.11. Western Blot Analysis

Naïve nerves and the distal stump (1.6 cm from the injured site) of IL nerves from vehicle- or BMMC-treated rats were dissected at different dpi and homogenized in TOTEX buffer containing protease inhibitor cocktail set III (Calbiochem-Milipore Sigma Chem Co. #535140, St Louis, MO, USA).

Proteins were quantified by means of Bradford’s method [78], electrophoretically separated on 12.5% SDS-PAGE, and transferred onto polvinylidine fluoride membranes for immunoblotting. Membranes were incubated with primary antibodies (overnight, 4 °C, Table 1a). After washing, membranes were incubated with horseradish peroxidase (HRP)-conjugated secondary antibodies (Table 1b) and developed using a chemiluminescent reagent (Biolumina, Kalium Technologies, Buenos Aires, Argentina) and the ImageQuant detector (LAS 500, GE Healthcare).

The relative intensity of myelin basic protein (MBP) and βIII-tubulin immunoreactive bands was analyzed using ImageJ software (National Institutes of Health, Bethesda, MD, USA) and normalized to GAPDH levels.

### 4.12. Preparation of Tissue Sections and Immunohistochemistry

IL and CL nerves from injured rats at different dpi were prepared for immunohistochemistry, following previously described methods [8,28]. Slices were incubated with primary antibodies (overnight, 4 °C, Table 1a) followed by incubation with secondary antibodies (Table 1b) plus DAPI (2 µg/mL, Sigma Chem Co, St. Louise, MO, USA). Tissues were mounted with Mowiol anti-fading solution for epifluorescence microscopy analysis using an Olympus BX100 epifluorescence microscope (Olympus, Tokio, Japan). Controls were performed by incubating the samples without the primary antibody [79,80].

### 4.13. Image Analysis

For migrating BMMC analysis, the number of recruited ^EGFP^BMMC in the distal stump was evaluated 7 days post-transplant, comparing between early and late transplant at different dpi; naïve animals were used as control. In immunofluorescence studies, at least 5 animals per experimental group were analyzed. Ten images from each condition were processed from regions 1.2–1.5 cm before crush (proximal near dorsal root ganglia (DRG)), 3–6 mm before crush (proximal near crush), or 3–6 mm after crush (distal) in each IL nerve.

In semi-thin sections, the total number of axons per 100 µm^2^ was calculated in 30 randomly selected fields. In ultra-thin sections, the total number of axons and those of small (<2 µm diameter) and large (≥2 µm diameter) caliber per 30 µm^2^ were calculated, as well as the number of myelin layers per 100 nm, the length of the intraperiod line, and the g-ratio (axon diameter/axon diameter including myelin sheath).

### 4.14. Statistical Analysis

All data were analyzed and quantified by experimenters who were blind to the experimental design. Statistical analysis was performed using GraphPad Prism (San Diego, CA, USA). Statistical tests, posttests, and significance values are indicated in figure legends. In all cases, α-value was set at 0.05.

## 5. Conclusions

We demonstrate a window of therapeutic opportunity and show that even late BMMC treatment in rats with transient peripheral nerve injury results in an immediate response to cell therapy, reflected in a gradual return to basal pain-like behavior and faster nerve regeneration in terms of morphology and functionality.

## Figures and Tables

**Figure 1 ijms-25-12482-f001:**
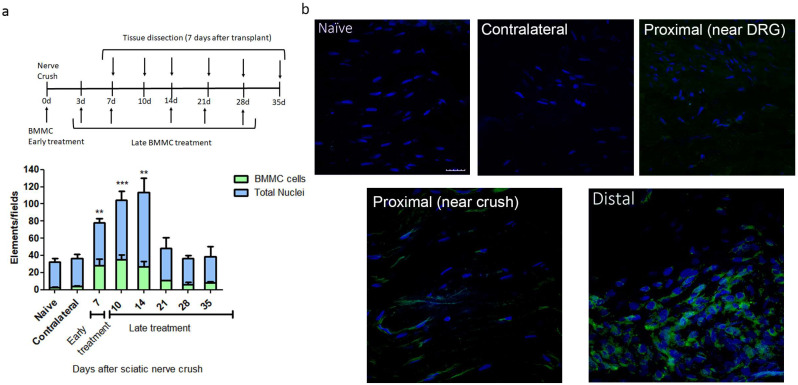
Kinetics of BMMC migration (*n* = 5, 60×, scale bar: 10 µm). (**a**) Experimental design indicating ^EGFP^BMMC transplant and tissue collection times. For all time-points, animals were sacrificed 7 days post-transplant: Bar graph showing total nuclei (blue) and ^EGFP^BMMC (green) per field in naïve and distal areas of IL nerves at different days post injury. Values are expressed as mean ± SD. Statistical analysis performed through one way-ANOVA followed by Dunnett’s post-hoc test. ** *p* < 0.01, *** *p* < 0.001 (^EGFP^BMMC on different days post injury vs. naïve). (**b**) Representative confocal microscopy images of a naïve, contralateral, proximal near DRG, proximal near crush and distal area of an ipsilateral nerve 14 days post injury showing DAPI-positive nuclei (blue) and transplanted ^EGFP^BMMC (green).

**Figure 2 ijms-25-12482-f002:**
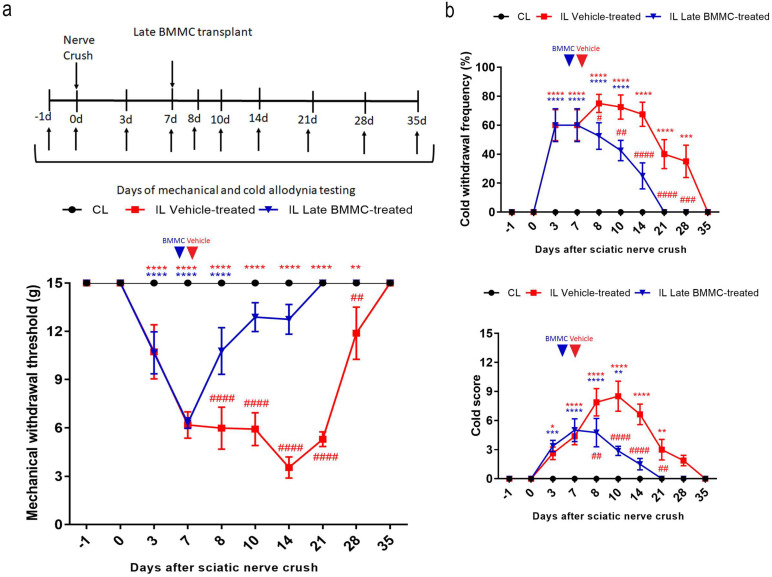
Effects of late BMMC transplant on mechanical and cold allodynia (*n* = 8). (**a**) Experimental design showing sciatic nerve crush, transplant, and pain-like behavioral test times. Mechanical withdrawal threshold calculated as 50% g threshold = (10 Xf + kd)/10,000, where Xf = value (in log units) of the final von Frey hair used; k = tabular value for the pattern of positive/negative responses; and d = mean difference (in log units) between stimuli. (**b**) Cold withdrawal frequency calculated as (N° of trials accompanied by brisk foot withdrawal) × 100/(N° of total trials); cold withdrawal score: 0 = no response, 1 = quick withdrawal, flick or stamp of the paw; 2 = prolonged withdrawal or repeated flicking (≥2) of the paw; 3 = repeated flicking of the paw with licking directed at the ventral side of the paw. Results correspond to contralateral (CL, black circle) pools data from vehicle-treated and late BMMC-treated rats and ipsilateral (IL) hind paws of vehicle-treated (red square) and late BMMC-treated rats (blue triangles). Values are expressed as mean ± SD. Blue arrows indicate the time-point of vehicle or BMMC administration. Statistical analysis performed through two-way ANOVA followed by Bonferroni post-hoc test. *p* < 0.0001 among groups in B and C; ## *p* < 0.001, ### *p* < 0.001, #### *p* < 0.0001 (IL vehicle-treated vs. IL late BMMC-treated); red * *p* < 0.05, ** *p* < 0.001, *** *p* < 0.001, **** *p* < 0.0001 (IL vehicle-treated vs. CL); blue ** *p* < 0.001, *** *p* < 0.001, **** *p* < 0.0001 (IL late BMMC-treated vs. CL).

**Figure 3 ijms-25-12482-f003:**
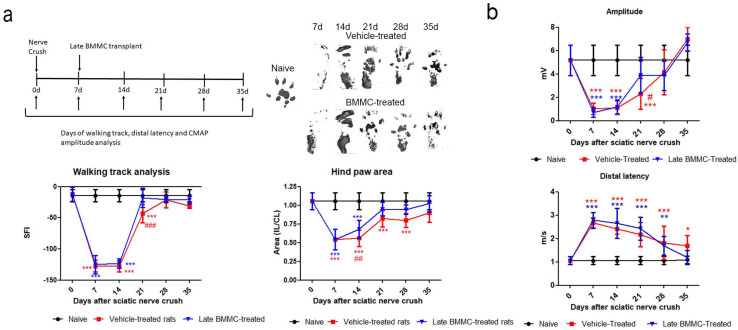
Effects of late BMMC transplant on SFI, distal latency, and CMAP (*n* = 6). (**a**) Experimental design showing sciatic nerve crush, transplant, and test times. Representative ipsilateral hind paw; sciatic functional index analysis (SFI) and ipsilateral/contralateral (IL/CL) hind paw area of naïve, vehicle- and BMMC-treated rats. (**b**) CMAP amplitude and distal latency of naïve, vehicle-and BMMC-treated rats. Values are expressed as mean ± SD. Statistical analysis performed through two-way ANOVA followed by Bonferroni post-hoc test. * *p* < 0.05; ** *p* < 0.01; *** *p* < 0.001 (vehicle-treated or BMMC-treated vs. naïve); # *p* < 0.05; ## *p* < 0.01; ### *p* < 0.001 (vehicle-treated vs. BMMC-treated).

**Figure 4 ijms-25-12482-f004:**
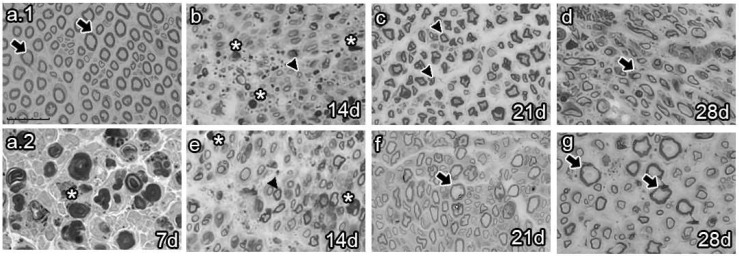
Semi-thin section images (*n* = 3, 40×, scale bar 20 µm). Images from semi-thin section of naïve rats (**a.1**), seven-days-post-sciatic nerve crush rats (**a.2**), vehicle-treated (**b**–**d**), and late BMMC-treated (**e**–**g**) rats at different survival times. Arrows indicate healthy axons, asterisks indicate myelin and axon debris, and arrowheads indicate newly formed axons.

**Figure 5 ijms-25-12482-f005:**
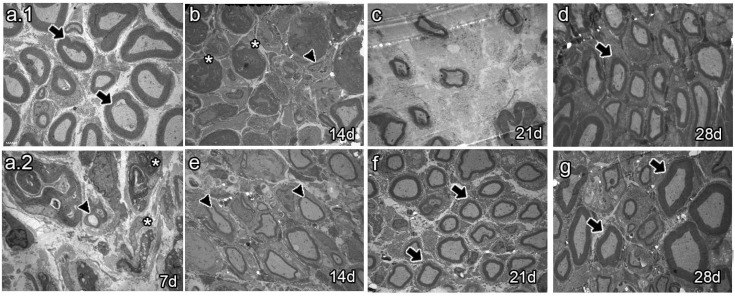
Ultrastructure images (*n* = 3, 3000×, scale bar 2 µm). Images from thin section, for ultrastructure analysis of naïve rats (**a.1**), seven-days-post-sciatic nerve crush rats (**a.2**), vehicle-treated (**b**–**d**), and late BMMC-treated (**e**–**g**) rats at different survival times. Arrows indicate healthy axons, asterisks indicate myelin and axon debris, and arrowheads indicate newly formed axons.

**Figure 6 ijms-25-12482-f006:**
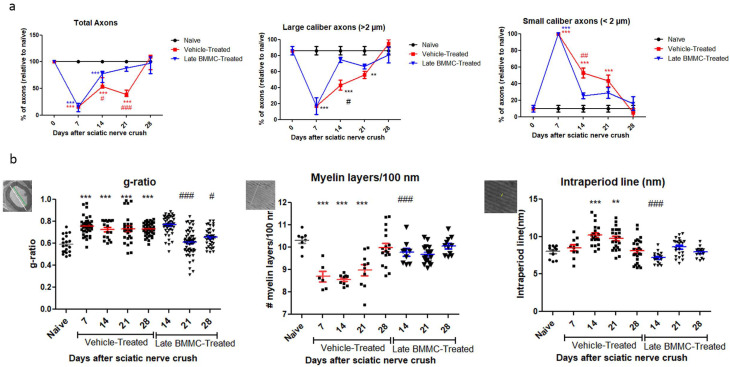
Ultrastructure quantification (*n* = 3): (**a**) Quantification of total axons/30 µm^2^, percentage of large- and small-caliber axons in vehicle-treated and late BMMC-treated rats at different dpi. Values are expressed as the mean ± SD. Statistical analysis was performed through two-way ANOVA followed by Bonferroni post-hoc test; ** *p* < 0.01; *** *p* < 0.001 (vehicle-treated or BMMC-treated vs. naïve rats); # *p* < 0.05; ## *p* < 0.01; ### *p* < 0.001 (vehicle-treated vs. BMMC-treated rats). (**b**) Quantification of g-ratio, myelin layers/100 nm, and intraperiod line length in vehicle-treated and late BMMC-treated rats at different dpi. Schematic images showing how each parameter in panel B was measured: for g-ratio, the green line represents axon diameter; the white line represents axon plus myelin diameter; for myelin layers, the line represents 100 nm and for intraperiod lines the line indicates it. Values are expressed as the mean ± SD. Statistical analysis was performed through one-way ANOVA followed by Bonferroni post-hoc test. ** *p* < 0.01; *** *p* < 0.001 (vehicle-treated or BMMC-treated vs. naïve rats), # *p* < 0.05; ### *p* < 0.001 (vehicle-treated vs. BMMC-treated rats).

**Figure 7 ijms-25-12482-f007:**
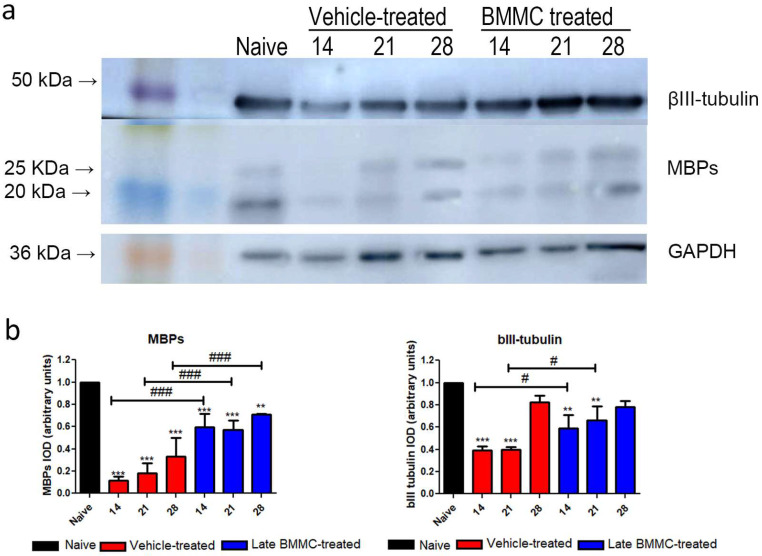
Immunoblot effects of late BMMC transplant on MBP and βIII-tubulin protein levels and distribution (*n* = 3). (**a**) Representative Western blot image of a naïve nerve, and the distal area of ipsilateral nerves of vehicle- and BMMC-treated rats for βIII-tubulin (~50 kDa) and MBP (~21 and 18.5 kDa). Rainbow molecular weight marker is shown on the left. (**b**) Quantification of MBP and βIII-tubulin integrated optical density (IOD) normalized to GAPDH (~36 kDa) levels and expressed relative to naïve nerves (AU, arbitrary units). In all cases, values are shown as mean ± SD. Statistical analysis performed through two-way ANOVA followed by Bonferroni post-hoc test. ** *p* < 0.01; *** *p* < 0.001 (vehicle-treated or BMMC-treated vs. naïve rats); # *p* < 0.05; ### *p* < 0.001 (vehicle-treated vs. BMMC-treated rats).

**Figure 8 ijms-25-12482-f008:**
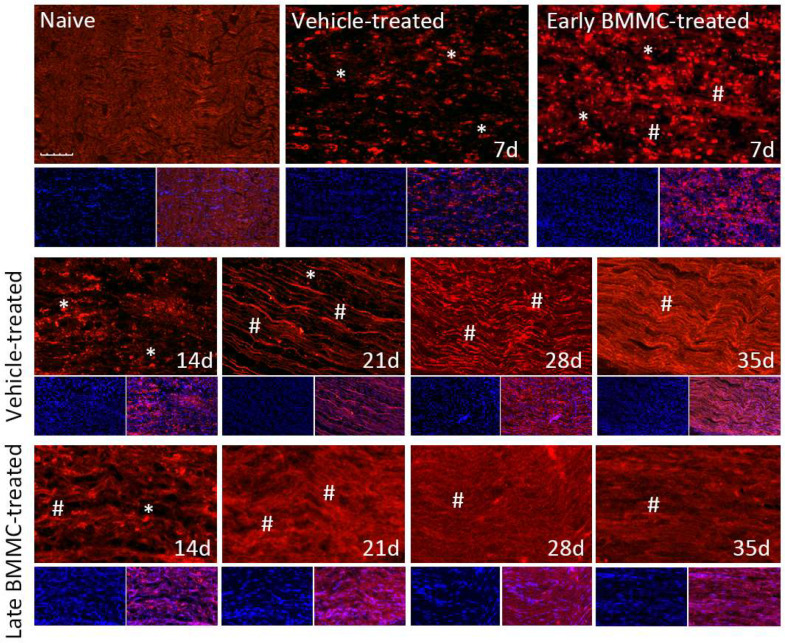
Immunohistochemical effects of late BMMC transplant on MBP-like immunoreactivity (*n* = 5, 20×, scale bar 50 µm). Longitudinal sections from naïve nerves and the distal area (3–6 mm after crush) of vehicle- and early-BMMC-treated animals analyzed 7 dpi. Middle and bottom panels show results obtained in the distal stump of vehicle- and late-BMMC-treated rats evaluated at different dpi. In blue, DAPI immunostaining for nuclei and in red MBP immunostaining. Asterisks indicate MBP clusters and hashtags indicate newly synthetized myelin.

**Figure 9 ijms-25-12482-f009:**
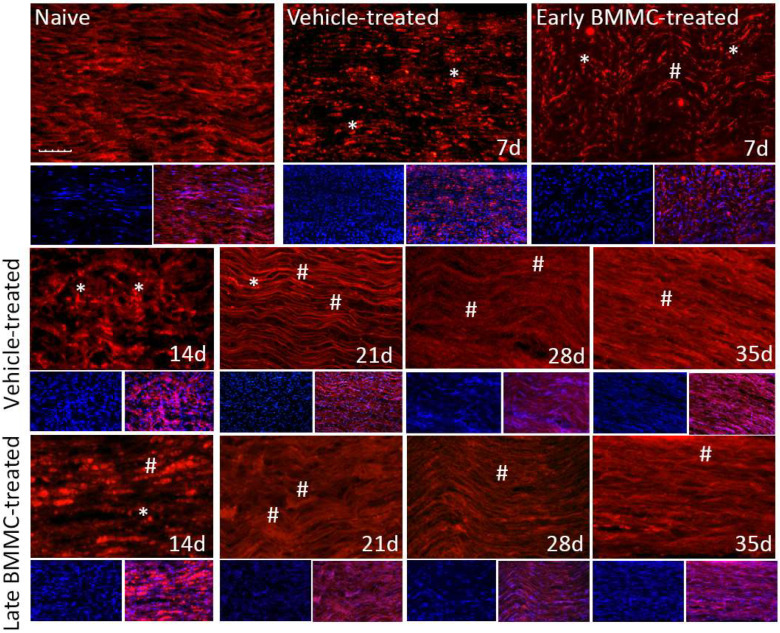
Immunohistochemical effects of late BMMC transplant on βIII-tubulin-like immunoreactivity (*n* = 5, 20×, scale bar 50 µm). Longitudinal sections from naïve nerves and the distal area (3–6 mm after crush) of vehicle- and early-BMMC-treated animals analyzed 7 dpi. Sections from the distal area of ipsilateral nerves of vehicle- and late BMMC-treated rats analyzed at different dpi. In blue is shown DAPI immunostaining for nuclei and, in red, βIII-tubulin immunostaining. Asterisks indicate βIII-tubulin clusters and hashtags indicate newly synthetized myelin.

**Figure 10 ijms-25-12482-f010:**
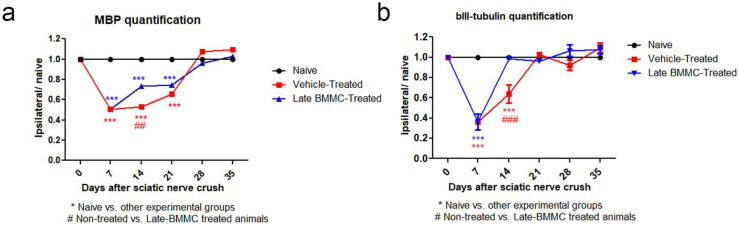
IOD quantification of MBP (**a**) and βIII-tubulin (**b**) (*n* = 5). IOD quantification shown as the mean ± SEM relative to naïve nerves. Statistical analysis performed through two-way ANOVA was followed by Bonferroni post-hoc test. *** *p* < 0.001 (naive vs. other experimental groups); ## *p* < 0.01; ### *p* < 0.001 (vehicle-treated vs. late BMMC-treated animals). Significance was set at *p* < 0.05.

**Figure 11 ijms-25-12482-f011:**
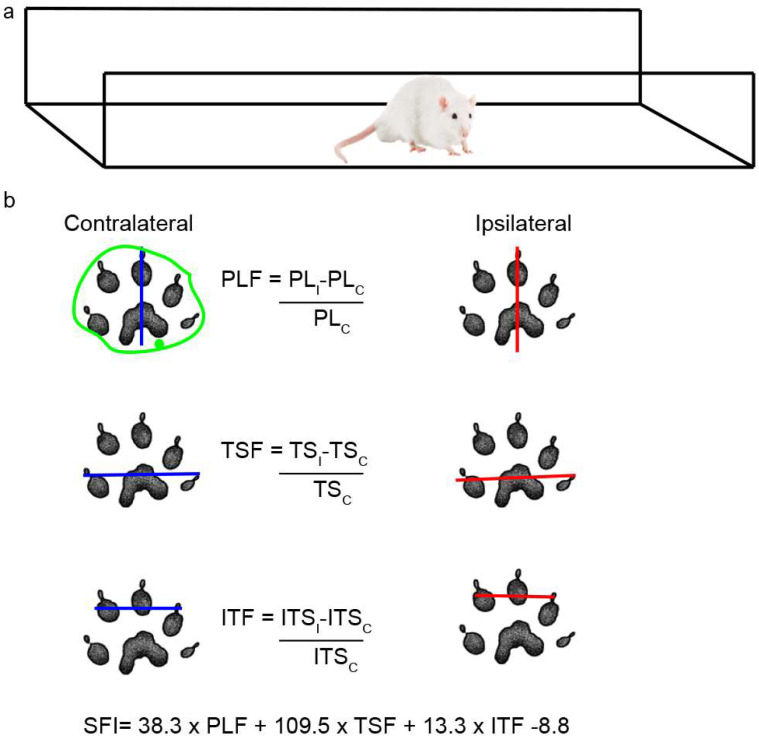
Scheme of walking track analysis. (**a**) Acrylic walkway (120 cm × 15 cm, ending in a darkened cage) used for walking track analysis. (**b**) In the footprint obtained, the length of the footprint from the third toe to the end of the foot (PL) and the distance between the first and fifth toe (TS) and between the second and fourth toes (ITS) were measured in both the CL and IL sides. The green outline shows an example of how each area was determined.

**Table 1 ijms-25-12482-t001:** (**a**) Primary antibodies used for WB and IHC. (**b**) Secondary antibodies used for WB and IHC.

Antigen	Cat #	Host	Clonality	Isotype (Clone)	Brand	Dilution	
						WB	IHC
(a) Primary antibodies							
MBP	800403	Mouse	Monoclonal	IgG_1_ (SMI99)	Biolegend *	1:2500	1:1500
βIII-tubulin	802001	Rabbit	Polyclonal	Poly18020	Biolegend *	1:7500	1:2500
*Loading control*							
GAPDH	Ab-8245	Mouse	Monoclonal	IgG_1_ (6C5)	Abcam **	1:5000	
(b) Secondary antibodies							
Reactivity	Cat #	Conjugate	Host	Brand		Dilution	
						WB	IHC
Mouse	115-035-146	HRP	Goat	Jackson ImmunoResearch ***		1:10,000	
Rabbit	111-035-003	HRP	Goat	Jackson ImmunoResearch ***		1:8000	
Mouse	A11030	Alexa546	Goat	Thermofisher ****			1:500
Rabbit	A21206	Alexa488	Donkey	Thermofisher ****			1:500

* Biolegend, San Diego, CA, USA; ** Abcam, Waltham, MA, USA; *** Jackson ImmunoResearch, West Grove, PA, USA; **** Thermofisher, Carlsbad, CA, USA.

## Data Availability

Data supporting these findings are available on reasonable request.

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
