# Peer review of "Late Bone Marrow Mononuclear Cell Transplantation in Rats with Sciatic Nerve Crush: Analysis of a Potential Therapeutic Time Window"

_ijms, 2024, doi:10.3390/ijms252312482_

Round 1

Reviewer 1 Report

Comments and Suggestions for Authors

In this study, the authors investigated whether there is a temporal window in which bone marrow mononuclear cells migrate more efficiently to damaged nerves while still retaining their positive effects. The results showed that transplanting BMMC between 0 and 7 days after injury led to the highest number of nested cells within the injured sciatic nerve. More importantly, administering BMMC seven days after a sciatic nerve crush was linked to the reversal of neuropathic pain, enhanced morphological appearance of the damaged nerves, and a trend towards faster recovery. While the study is generally interesting, there are several minor points that should be addressed before publication.

1. Why were only male rats used in the behavioral studies? What was the reason of this?

2. The style of scale bars and graphs is not uniform. Please correct it.

3. The captions for Figures 4. and 5. are difficult to understand.

4. The scale bars in Figure 8. And 9. are unreasonably large.

Author Response

Comments 1: Why were only male rats used in the behavioral studies? What was the reason of
this?

Response 1: We thank the Reviewer for this question, pointing out at one limitation of our
present manuscript. In fact, a number of studies over recent years have exposed and analyzed
a variety of sex-related differences in the physiopathology of chronic pain (Sorge et al., 2015;
Tave et al., 2016; Gregus et al., 2021; Lopes et al., 2017; Mogil 2020). The results in the
manuscript presented here are part of a larger project initially developed in male rats. While
several aspects of this project have already been published (Usach et al., 2011, 2017; Piñero et
al., 2018, 2023), the current manuscript represents our final aim at defining the temporal window
at which the beneficial effects on pain behavior of BMMCs and some of the mechanisms
involved can be harnessed, also in male rats. We are, however, already working on determining
if male-vs-female differences exist in the responses to BMMC administration, using a different
experimental model of sciatic nerve injury, and hope to publish this data in the near future to
provide this critical piece of information. We have now included in the Discussion section a
paragraph, in order to address this limitation (page 11 and 12, lines 327-332).
- Sorge RE, Mapplebeck JC, Rosen S, Beggs S, Taves S, et al. 2015. Different immune cells
mediatemechanical pain hypersensitivity in male and female mice. Nat. Neurosci. 18:1081–83
-Taves S, Berta T, Liu DL, Gan S, Chen G, et al. 2016. Spinal inhibition of p38 MAP kinase
reducesinflammatory and neuropathic pain in male but not female mice: sex-dependent
microglial signaling inthe spinal cord. Brain Behav. Immunity 55:70–81
- Gregus AM, Levine IS, Eddinger KA, Yaksh TL, Buczynski MW. Sex differences in
neuroimmune and glial mechanisms of pain. PAIN 2021; 162:2186–200
- Lopes DM, Malek N, Edye M, Jager SB, McMurray S, McMahon SB, Denk F. Sex differences
in peripheral not central immune responses to pain-inducing injury. Sci Rep 2017; 7:16460.
- Mogil JS. Qualitative sex differences in pain processing: emerging evidence of a biased
literature. Nat Rev Neurosci2020; 21:353–65

Comments 2: The style of scale bars and graphs is not uniform. Please correct it

Response 2: Following reviewer suggestions all scale bars and graph are now in the same
format.

Comments 3: The captions for Figures 4. and 5. are difficult to understand.

Response 3: Following reviewer suggestions caption for figure 4 (page 5, lines 159-162) and
caption for figure 5 (page 6, lines 174-177) were changed to a new version.

Comments 4: The scale bars in Figure 8. And 9. are unreasonably large.
Response 4: We thank the reviewer observation, as there was a mistake adding the scale bar in
those figures. The correct one is now place in each figure.

Reviewer 2 Report

Comments and Suggestions for Authors

The manuscript entitled „Late bone marrow mononuclear cell transplantation in rats with sciatic nerve crush. Analysis of potential therapeutic time window.“ By Vanina usach et al is of interest for the readers of the International Journal of Molecular Sciences.

The authors analysed the effect of BMMC derived from donor rats on functional parameters of nerve generation after sciatic nerve crush. The main focus was set on the ideal timing of cell application after nerve injury. Two time points were analysed, 7 days and 14 days after injury. Cell transplantation after 7 days lead to a significant increase of BMMC in the surrounding of the crushed nerve and healing parameters improved significantly within a short period of time. If cells were provided later, surprisingly, also an healing reaction was observed but much less cells were recruited to the injury site and also improvement of nerve function occured with latency.

The manuscript is original as proved with Pubmed and google scholar research. It is well written, generally the data adequately presented and discussed. However some issues needs further clarification as indicated in the following.

1.)    Experimental groups: Please provide a table with group setup and numbers of animals for each group and experiment.

2.)    In which volume BMC were applied? Please add this information.

3.)    BMMC consist of several different cell populations and the amount of stem cells is rather low. There is evidence, at least from the area of bone defect healing that stem cells (hematopoietic stem cells) within BMMC play only a minor role for the healing whereas the monocyte subpopulation has a significant impact.

Do you have data which cell types ccumulate at the site of nerve injury? If samples were left please provide some additional immuno fluorescence analysis which cell type(s) preferentially accumulate at the injured site (e.g. F4/80 for macrophages, CD34 for HSC, concomittant to the EGF-expression). If this is not possible please expand the corresponding part of the discussion.

Author Response

Comments 1: Experimental groups: Please provide a table with group setup and numbers of
animals for each group and experiment.

Response 1:

Animal
group

Kinetics
of BMMC
migration

Behavioral

studies

Electrophysiology

studies

Semi     and
ultrathin
analysis

Western

Blot

Immunofluorescence

Naive 5 14 30 3 3 5
Early BMMC-treated 5 - - - - 5
Vehicle-treated - 14 30 12 9 25
Late BMMC-treated 25 14 30 9 9 20

Table indicating the total number of animals from each group used in the different experimental
procedures, the number of animals per survival times analyzed in each experimental procedure
are indicated in the corresponding figure legend.
Whether the reviewer consider it necessary this table may be included in the Material and
Methods section.

Comments 2: In which volume BMC were applied? Please add this information.

Response 2: Considering reviewer comment, the volume in which 1.10 7 BMMC were transplanted
through the lateral tail vein was added in the Materials and Methods section (page 13, line 381)

Comments 3: BMMC consist of several different cell populations and the amount of stem cells is
rather low. There is evidence, at least from the area of bone defect healing that stem cells
(hematopoietic stem cells) within BMMC play only a minor role for the healing whereas the
monocyte subpopulation has a significant impact.
Do you have data which cell types ccumulate at the site of nerve injury? If samples were left please
provide some additional immuno fluorescence analysis which cell type(s) preferentially accumulate
at the injured site (e.g. F4/80 for macrophages, CD34 for HSC, concomittant to the EGF-
expression). If this is not possible please expand the corresponding part of the discussion.

Response 3: We thank the Reviewer for this insightful observation. Unfortunately, we do not have
enough remaining material to perform additional immunofluorescence analysis. However, it may be
worth mentioning that we have already published an intensive characterization of BMMCs previous
to their transplant, showing that they are enriched in blast, lymphocytes and monocytes (Usach et
al., 2017, Piñero et al., 2018). On the other hand, only a small percentage of transplanted cells
appears to be integrated in the injured tissue, where they undergo transdifferentiation to Schwann
cells, macrophages and axonal cells, and losing CD34 expression (Usach et al., 2011), altogether
making it difficult to evaluate their original phenotype in vivo. However, and as the Reviewer points
out, it has already been proposed that the therapeutic potential of BMMCs lies on the heterogeneity

of cells in the samples, and their capacity to exert immunomodulating influences, promoting the
release of anti-inflammatory cytokines, reducing the synthesis and release of pro-inflammatory
cytokines, and facilitating regeneration and neuropathic pain relief (Piñero et al., 2023).
Following reviewers´suggestion a paragraph detailing the subpopulations of BMMC and their
characterization in vivo, was added to the discussion section (page 11, lines 309-317)